# `WalkLM`: A Uniform Language Model Fine-tuning Framework for Attributed Graph Embedding

**Yanchao Tan**
College of Computer and
Data Science
Fuzhou University
Fuzhou, China
yctan@fzu.edu.cn

**Zihao Zhou**
College of Computer and
Data Science
Fuzhou University
Fuzhou, China
reviverkey@gmail.com

**Hang Lv**
College of Computer and
Data Science
Fuzhou University
Fuzhou, China
lvhangkenn@gmail.com

**Weiming Liu**
College of Computer Science
Zhejiang University
Hangzhou, China
21831010@zju.edu.cn

**Carl Yang**[*]
Department of Computer Science
Emory University
Atlanta, United States
j.carlyang@emory.edu

## Abstract

Graphs are widely used to model interconnected entities and improve downstream predictions in various real-world applications. However, real-world graphs nowadays are often associated with complex attributes on multiple types of nodes and even links that are hard to model uniformly, while the widely used graph neural networks (GNNs) often require sufficient training toward specific downstream predictions to achieve strong performance. In this work, we take a fundamentally different approach than GNNs, to simultaneously achieve deep joint modeling of complex attributes and flexible structures of real-world graphs and obtain unsupervised generic graph representations that are not limited to specific downstream predictions. Our framework, built on a natural integration of language models (LMs) and random walks (RWs), is straightforward, powerful and data-efficient. Specifically, we first perform attributed RWs on the graph and design an automated program to compose roughly meaningful textual sequences directly from the attributed RWs; then we fine-tune an LM using the RW-based textual sequences and extract embedding vectors from the LM, which encapsulates both attribute semantics and graph structures. In our experiments, we evaluate the learned node embeddings towards different downstream prediction tasks on multiple real-world attributed graph datasets and observe significant improvements over a comprehensive set of state-of-the-art unsupervised node embedding methods. We believe this work opens a door for more sophisticated technical designs and empirical evaluations toward the leverage of LMs for the modeling of real-world graphs.

## 1 Introduction

Graphs are widely used to model interconnected entities, and they are critical in enhancing downstream predictions in various real-world applications. Nowadays, real-world graphs are often associated with complex attributes on multiple types of nodes and even links [27, 28], and modeling such real-world graphs is non-trivial. For example, in the schema of a clinical attributed graph constructed

---

[*]Corresponding author

37th Conference on Neural Information Processing Systems (NeurIPS 2023).

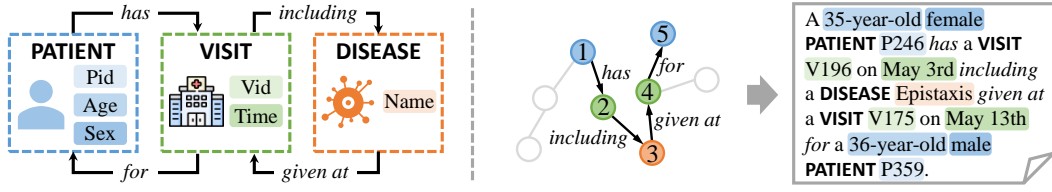

|  (a) Schema of real-world attributed graph | (b) Attributed random walk | (c) Composed text |

Figure 1: **A toy example of the transformation from a real-world attributed graph to the composed text**. (a) is a schema of a real-world attributed graph on MIMIC-III that delineates how nodes (e.g, patients), edges (e.g, *has* between patients and visits), and the associated attributes (e.g., age) are organized and interconnected. (b) is attributed random walk for capturing structural information and can be composed to text in (c).

from MIMIC-III[2] data (shown in Figure 1(a)), there are multiple types of nodes such as patients, visits and diseases, each with their own attributes such as age, sex, time and name; there are also multiple types of links associated with different meanings. Such complex heterogeneous attributes of nodes and links attributes can hardly be modeled in a uniform space.

Although the widely used graph neural networks (GNNs) have shown remarkable successes in the modeling of attributed graphs for various downstream applications [5, 14, 53, 53], the representation learning (a.k.a. embedding) of GNNs often requires sufficient training toward specific downstream predictions to achieve strong performance [4, 76]. While unsupervised training has also been explored for GNNs [23, 63], the generic performances of unsupervised GNN embeddings across different downstream tasks are still unsatisfactory (as we will also show in our experiments). Consequently, it is important to devise a general-purpose graph embedding method to simultaneously understand the complex node/link attributes and incorporate the flexible graph structures in an unsupervised manner.

However, two obstacles stand in the way of achieving this goal. First, the nature of the attributes can be intricate and diverse, thus understanding their semantics in a uniform space is non-trivial. Second, the graph structures need to be accurately captured and incorporated into the embedding space, which is not straightforward due to the inherent flexibility and potential complexity of entity relations.

To address these issues, in this work, we take a fundamentally different approach than GNNs, named `WalkLM`, which is a uniform framework to obtain unsupervised generic graph representations that are not limited to specific downstream tasks. To this end, we first draw inspiration from the recent successes of language models (LMs), and propose to leverage LMs as uniform attribute models that can capture the intricate semantics of complex heterogeneous attributes of nodes and links. Secondly, we propose to leverage the classic tool of random walks (RWs) on graphs which have been shown effective in capturing flexible graph topological structures by various studies [8, 10, 13, 21, 38].

Specifically, we first generate attributed RWs on the graph (e.g., $1 \xrightarrow{has} 2 \ldots \xrightarrow{for} 5$ in Figure 1(b)), and design an automated textualization program to compose roughly meaningful textual sequences directly from the attributed RWs. As shown in Figure 1(c), the composed text is a mapping from the attributed RW in Figure 1(b), where a uniform automatic program firstly textualize different types of nodes (in different colors) by properly concatenating the nodes with the names and values of different attributes, and then textualize the whole RWs by concatenating the nodes and links. Furthermore, we fine-tune an LM using the RW-based textual sequences and extract embedding vectors from the LM. The learned embeddings encapsulate both attribute semantics and graph structures, and can be flexibly utilized for arbitrary downstream tasks.

In our experiments, we take the node embeddings and evaluate them towards different downstream prediction tasks (e.g., node classification and link prediction) on multiple real-world attributed graph datasets and observe significant improvements over a comprehensive set of state-of-the-art unsupervised node embedding methods (e.g., `WalkLM` achieves an average of 88.98% improvement over the state-of-the-art baselines ranging from existing RW-based graph embedding methods to popular unsupervised GNN modes regarding both Macro-F1 and Micro-F1 metrics). We believe this

---

[2]https://physionet.org/content/mimiciii/1.4/

work paves the way for more sophisticated technical designs and empirical evaluations toward the leverage of LMs for the modeling of real-world graphs.

## 2 Related Work

**Graph Representation Learning.** In recent years, a plethora of representation learning techniques are proposed for graphs [3, 15, 55, 58, 74]. In this work, we focus on the objective of learning embedding vectors for each node that characterizes its topological (and semantic) information in the graph. Among existing node embedding methods, many have analyzed and utilized the great promise of random walks (RWs) in capturing the topological structures of graphs [8, 13, 21, 38]. However, the above methods ignore the abundant attribute information surrounding the nodes and edges [32]. In recent studies, Graph neural networks (GNNs) for learning node representations through aggregating information from neighboring nodes on graphs [14, 24, 62]. However, most existing GNNs are established in a supervised learning setting, which requires abundant task-specific labeled data that may not be available in real-world applications [4, 76], and the embeddings they learn are not generalizable across different downstream tasks [70]. Although some studies tried to reduce the labeling effort by pre-training an expressive GNN model on unlabeled data with self-supervision methods (e.g., contrastive learning) [19, 22, 75], their performances in specific downstream tasks still rely much on the properly chosen self-supervision tasks and attribute encoders [47, 73]– that is, there still lack a uniform framework for generic unsupervised attributed graph representation learning.

**Language Models (LMs).** With the massive corpora and powerful computation resources for pre-training, modern language models (LMs) derive various families [34]. These LMs can be grouped into: (1) auto-regressive LMs (e.g., GPT [40] and GPT-2/3 [1, 41]), (2) masked LMs (e.g., BERT [7], RoBERTa [31], and XLNet [64]), and (3) encoder-decoder LMs (e.g., BART [26] and T5 [42]). LMs have been intensively studied by NLP researchers for various language-related tasks [16, 29, 43]. In our work, we innovatively utilize LMs as uniform attribute models for nodes and links in graphs for the first time. Note that, our work also readily generalizes to recent large language models (LLMs) (e.g., InstructGPT [36], ChatGPT and GPT-4 [35]) via appropriate parameter-efficient training approaches (e.g., LoRA [18] and prompt-tuning [25, 30]). However, those are orthogonal to the innovations in this work, for which we leave the exploration as immediate future works.

**LMs with Knowledge Graph (KG).** In recent studies, combining LMs with KG has been widely applied in various real-world applications [37]. Among existing methods, many have proposed to enhance LMs with KG for significantly improve the performance of LMs in accessing domain-specific knowledge [46, 56, 67, 69], and the others proposed to harness the power of LMs for addressing KG-related tasks [65, 66]. However, the above methods fail to effectively combine the rich semantic information of graphs with global topological information. Furthermore, as closest to us, [12] proposed to combine random walks and neural network language models to produce new word representations. However, it ignores the rich relational information between nodes and thus fails to learn richer topological information. Moreover, compared with modern LMs with extensive prior knowledge, the text-based encoder used in [12, 51] fails to extract richer semantic information.

## 3 Preliminaries

### 3.1 Problem Formulation

Given an attributed graph $\mathcal{G}$ and multiple downstream tasks $\tau_i$ (e.g., node classification $\tau_1$ and link prediction $\tau_2$), the goal of WalkLM is to sufficiently model information in $\mathcal{G}$ and improve task performances on $\tau_i$. Specifically, we denote the graph as $\mathcal{G} = (V, E, \Phi, \Psi)$, where each node $v_i \in V$ is associated with attributes $\Phi(v_i)$, and each edge $e_i \in E$ is associated with attributes $\Psi(e_i)$.

To fully exploit both the semantic information in $\Phi$ and $\Psi$, and the structural information in $V$ and $E$, we first design attributed random walks (RWs) based automated textualization program on $\mathcal{G}$, where we can transform the attributed graph to the meaningful textual sequences $\mathbb{W} = \{W_i\}_{i=1}^N$. Then, we fine-tune a graph-aware language model (LM) using the RW-based textual sequences $\mathbb{W}$, find the optimized parameters $\Theta$ of the LM, and extract embedding vectors from the LM. Finally, we apply these embeddings to $\tau_1$ and $\tau_2$ for performance evaluation.

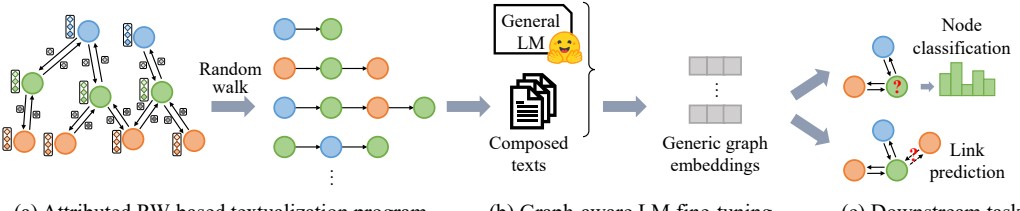

| (a) Attributed RW-based textualization program | (b) Graph-aware LM fine-tuning | (c) Downstream tasks |

Figure 2: **The overall framework of** `WalkLM`**:** (a) An attributed random walk (RW) based textualization program obtains attributed RWs and composes roughly meaningful textual sequences directly from the attributed RWs. (b) A pre-trained general LM is fine-tuned in a graph-aware fashion using the generated textual sequences to produce generic graph embeddings. (c) The learned graph embeddings can be applied to various downstream tasks.

## 3.2 Masked Language Modeling

Masked Language Modeling (MLM) is a widely-used and highly effective technique for training large-scale language models [7]. In the MLM task, 15% of the language tokens are randomly chosen for prediction. If the $i$-th token in a sequence $W$ is chosen, it can be replaced by (1) the [MASK] token 80% of the time, (2) a random token 10% of the time, and (3) the unchanged $i$-th token 10% of the time. Then, token $t_i$ will be used to predict the original token with following cross-entropy loss [7, 59, 60]:

$$\mathcal{L}_{MLM} = -\frac{1}{|\mathbb{X}|} \sum_{X \in \mathbb{X}} \sum_{t_i \in \mathcal{M}} \log p\left(t_i | T_{\backslash i}\right), \tag{1}$$

where $\mathbb{X}$ is a set of training examples, $\mathcal{M}$ is the prediction set of the masked token, $T_{\backslash i} = \{t_1, \ldots, t_{i-1}, t_{i+1}, \ldots, t_L\}$ is the surrounding token set of $t_i$, and $|T_{\backslash i}| = L - 1$.

## 4 Methodology

In this section, we present the proposed method `WalkLM`, which comprises two major components. The first component, the attributed random walk (RW) based textualization program, captures both topological structures and attribute information of the graph and composes corresponding text automatically. The second component, the graph-aware language model (LM) fine-tuning, leverages pre-trained LM to encode the complex semantics along with the graph structure. The overall model architecture is shown in Figure 2.

### 4.1 Attributed RW-based Textualization Program

To model node/link attributes, traditional machine learning algorithms require a standard process of *vectorization*, which transforms different types of attributes into categorical or numerical features as model input. However, such a vectorization process removes the actual semantic meanings of the attributes, and it cannot unify different types of attributes (e.g., ages, sexes, time, etc.) in the same space. Inspired by the recent successes of LMs, we find it promising to leverage pre-trained LMs to understand the intricate semantics of complex heterogeneous attributes [29, 34]. The key idea is to perform *textualization* instead of vectorization, that is, to transform different types of attributes into texts, which can then be modeled by the LMs in a uniform space. Therefore, we first design the following process for the textualization of individual entities in the graph (i.e., nodes and edges).

**Entity-level Textualization.** Inspired by a wide range of NLP tasks that leverage prompts to construct informative rule templates, we propose to textualize attributed graph entities via a rule-based program function $\mathcal{P}(\cdot)$ that automatically concatenates attribute values with the corresponding attribute names, as well as attributes and the corresponding entity id. For example, as shown in Figure 1, for a patient node $v_i$ with attributes $\Phi(v_i) = \{age : 35, \; sex : \text{female}, \; pid : \text{P246}\}$, the texualization program will convert it into $\mathcal{P}(v_i) = < \text{A 35-year-old female patient P246} >$. For edges, in this work, we only consider simple relational edges such as *has* and *including*, which are already texts, but our framework is readily extensible to edges with more complex attributes.

For attributed graphs, after modeling the complex attributes of individual entities, the next challenge would be to model the flexible graph topological structures. To this end, we propose to utilize the

powerful and efficient tool of RWs, as the foundation for the textualization of graphs. Specifically, we design the following process:

**Walk-level Textualization.** We first initiate an attributed RW $W$ by randomly selecting a node $v_0$ and attaching the textualized node information $\mathcal{P}(v_0)$ to $W$. Then, we extend $W$ by randomly selecting an edge $e_1$ starting from $v_0$ as in a standard RW with a uniform probability as 1 divided by the number of out-edges of $v_0$, with its terminating node $v_1$, and appending the corresponding texts $\mathcal{P}(e_1)$ and $\mathcal{P}(v_1)$ to $W$. We can keep adding edges and nodes until the random walk terminates, such as based on a termination probability $\alpha$. The final textual sequence $W = \{\mathcal{P}(v_0), \mathcal{P}(e_1), \mathcal{P}(v_1), \ldots, \mathcal{P}(v_{L-1}), \mathcal{P}(e_L), \mathcal{P}(v_L)\}$, which corresponds to an actual attributed random walk with the length of $2L + 1$ on the graph, will be a roughly meaningful sentence, such as the one shown in Figure 1(c).

After performing the RW for $N$ times, we can obtain $N$ attributed RWs as $\mathbb{W} = \{W_i\}_{i=1}^N$, which constitutes a graph-aware corpus for training LMs without any downstream task supervision.

**Discussion.** We believe that the above proposed attributed RW-based textualization program is effective in capturing both complex attribute information and flexible topological structures of the graph in an unsupervised manner. Such ability arises from two critical properties: (1) Random walks are known to be capable of providing characteristic graph traits and reconstructing network proximity of nodes [21, 33]. To be specific, it has been proven that the distribution of random walks starting at a particular node, which can be estimated with sufficient numbers of random walks, can sufficiently preserve the subgraph structure around the node. This means a sufficient amount of attributed RWs from different nodes can well reflect the topological structures of graphs. (2) Our textualization program completely preserves the attributes of nodes and edges, as well as the whole RWs, and it presents such information as meaningful texts, which is natural for LMs to comprehend. Moreover, RWs are known to be efficient and highly parallelizable, where numerous threads can run simultaneously to generate large amounts of RWs [9]. Note that, we only need to perform the rule-based textualization once for every node during the pre-processing stage, which is also efficient and highly amenable to parallelization.

### 4.2 Graph-Aware LM Fine-Tuning

Despite the robust generalizability of LMs, fine-tuning remains a necessary step [57, 68], which allows the general LM to adapt its broad language capabilities to the specificities of the different attributed graphs.

As one of the mainstream language modeling techniques, masked language modeling (MLM) is proven to sufficiently utilize textual semantic information for further fine-tuning LMs [39, 77]. To achieve the balance between effectiveness and efficiency, we propose a graph-aware LM fine-tuning mechanism with knowledge distillation [2, 17, 44]. Specifically, we adopt a general LM DistilRoBERTa (abbr. DRoBERTa)[3] as our starting point for fine-tuning, where RoBERTa is a widely used successor of BERT [7]. Note that, DRoBERTa can further reduce the size of the original RoBERTa model by 40% and achieve 60% faster training while retaining 97% capacity of RoBERTa's language understanding [44]. Then, we feed the attributed RW $W \in \mathbb{W}$ to the LM tokenizer and obtain the corresponding token list $\mathcal{T} = \{t_1, t_2, \ldots, t_K, <\text{MASK}>_1, <\text{MASK}>_2, \ldots, <\text{MASK}>_{|\mathcal{M}|}\}$, where $t_i$ denotes the unmasked token, and $<\text{MASK}>_i$ denotes the token chosen for prediction. In this way, we create a training example $X_i = \langle W, \mathcal{T} \rangle \in \mathbb{X}$ for fine-tuning, where $\mathbb{X} = \{X_1, X_2, \ldots, X_N\}$ is the whole training dataset. We adopt the cross-entropy loss as the fine-tuning objective [7, 44, 59], which is formulated as follows:

$$\mathcal{L}_{FT}(\Theta) = -\frac{1}{|\mathbb{X}|} \sum_{X_i \in \mathbb{X}} \left[ \sum_{t_k^* \in \mathcal{M}} \log \frac{\exp(Sim(t_k, t_k^*))}{\sum_{t \in \mathcal{V}} \exp(Sim(t_k, t))} \right], \tag{2}$$

where $\Theta$ is the learnable parameters of our graph-aware LM, $\mathcal{M}$ is the ground-truth set of the masked token, $\mathcal{V}$ is the token vocabulary, $t_k$ is the prediction token, $t_k^*$ is the ground-truth token, and $Sim(t_i, t_j)$ is the similarity scoring function between $t_i$ and $t_j$.

After obtaining the fine-tuned LM based on MLM, we can extract generic graph embeddings (e.g., node embeddings based on node name). For example, we can access the representation of disease embedding in Figure 1(c) via extracting the embedding of $\text{Epistaxis}$.

---

[3]https://github.com/huggingface/transformers/tree/main/examples/research_projects/distillation

**Complexity Analysis.** The time complexity of fine-tuning is $\mathcal{O}(Iter \cdot |N| \cdot l_{avg}^2 \cdot d)$, where $Iter$ is the number of iterations of training, $N$ is the number of training examples, $l_{avg}$ is the average length of input textual sequences for the LM and $d$ is the dimension of embedding. We infuse global graph structure knowledge into the LM to distinguish similar positions instead of negative sampling, making it possible to fine-tune more efficiently [59].

**Model Extension.** Our framework is a fundamental approach to integrate LMs and RWs for generic attributed graph embedding, which can choose different LMs according to different tasks and domains (shown in our experiments in Sec. 5.4) and generalize to recent large language models (LLMs) (e.g., InstructGPT [36], ChatGPT and GPT-4 [35]) via appropriate parameter-efficient training approaches such as LoRA [18]. A full exploration of different LMs is orthogonal to the main contributions in this work, which is left as future work.

**Datasets Extension.** Our method introduces the novel process of textualization, which converts general attributed graphs into text-like sequence data. This process allows us to leverage the capabilities of pre-trained language models for graph representation learning. Note that, our method only requires some meaningful attributes on the graphs, which are available in most real-world graphs such as biological networks, social networks, and knowledge graphs. Some preliminary experimental results of graph classification and KG-related datasets are shown in Appendix A.1 and Appendix A.2.

### 4.3 Various Downstream Tasks

In this work, we focus on node embeddings since they are most commonly studied for graph representation learning, and it is straightforward to extract node embeddings from the fine-tuned LM based on node names (or node IDs if the node has no meaningful name). However, `WalkLM` can also easily generate edge embeddings, by adding edge names (e.g., relation names) or edge IDs to the textualization process, and even obtain subgraph/graph embeddings via appropriate embedding aggregation mechanisms. The extracted embeddings can be directly used as fixed feature vectors to train downstream prediction models for tasks such as node classification or link prediction. Alternatively, these embeddings can also serve as initialization for more learnable embeddings in complex neural network models, which can be further updated according to the specific requirements of the downstream task.

## 5 Experiments

### 5.1 Experimental Setup

**Datasets.** We conduct extensive experiments on two real-world datasets, PubMed [4] and MIMIC-III [5]. PubMed contains a graph of genes, diseases, chemicals, and species. The nodes and edges are extracted according to [61]. A relatively small fraction of diseases are grouped into eight categories. MIMIC-III contains a graph of diseases, patients, and visits, where nodes and relations are extracted from clinical records. Diseases are classified into nineteen categories according to ICD-9-CM [6]. The detailed statistics are shown in Table 1.

Table 1: Statistics of the datasets.

| Dataset | #attribute type | #node type | #node | #link type | # link | #label | #label node |
|---------|-----------------|------------|-------|------------|--------|--------|-------------|
| PubMed | 8 | 4 | 63,109 | 10 | 244,986 | 8 | 454 |
| MIMIC-III | 10 | 3 | 32,267 | 4 | 559,290 | 19 | 4880 |

**Competitors.** We compare our proposed `WalkLM` with ten graph-oriented baselines that are designed for heterogeneous information networks (HINs) or knowledge graphs (KG), which can handle different types of nodes and edges. We divided them into four groups as follows:

(1) RW-based methods: **Metapath2Vec** (abbr. M2V) [8] proposes to use user-defined meta-paths as guidance, so as to learn node embeddings on HINs. **HIN2Vec** [10] carries out multiple prediction

---

[4]https://pubmed.ncbi.nlm.nih.gov/

[5]https://physionet.org/content/mimiciii/1.4/

[6]https://www.cdc.gov/nchs/icd/icd9cm.htm

Table 2: Different downstream task results (%) with the corresponding std (±) on two datasets. The best performances are in bold and the second runners are shaded in gray, where * denotes a significant improvement according to the Wilcoxon signed-rank significance test.

| Task | Node Classification | | | | Link Prediction | | | |
|---|---|---|---|---|---|---|---|---|
| Dataset | PubMed | | MIMIC-III | | PubMed | | MIMIC-III | |
| Metric | Macro-F1 | Micro-F1 | Macro-F1 | Micro-F1 | AUC | MRR | AUC | MRR |
| M2V | 15.35 | 20.27 | 19.69 | 29.24 | 74.53 | 89.58 | 75.05 | 88.32 |
| | (±1.27) | (±3.01) | (±0.62) | (±1.57) | (±3.79) | (±2.05) | (±0.41) | (±0.23) |
| HIN2Vec | 11.57 | 18.92 | 19.12 | 28.05 | 74.21 | 90.56 | 73.46 | 88.10 |
| | (±1.23) | (±2.78) | (±1.32) | (±1.44) | (±5.49) | (±1.06) | (±0.41) | (±0.14) |
| ConvE | 16.06 | 19.16 | 24.44 | 32.89 | 76.48 | 92.27 | 69.56 | 84.88 |
| | (±3.69) | (±4.00) | (±1.28) | (±0.86) | (±4.31) | (±0.57) | (±0.36) | (±0.25) |
| ComplEx | 13.93 | 18.27 | 9.82 | 21.39 | 79.81 | 91.79 | 63.86 | 81.40 |
| | (±2.59) | (±4.12) | (±0.56) | (±3.12) | (±0.97) | (±0.48) | (±0.42) | (±0.40) |
| SimKGC | 21.97 | 30.83 | 51.62 | 58.50 | 79.62 | 91.43 | 67.73 | 84.86 |
| | (±3.51) | (±3.10) | (±1.81) | (±1.52) | (±2.72) | (±0.48) | (±1.69) | (±0.54) |
| RGCN | 12.50 | 18.50 | 7.19 | 14.55 | 72.08 | 88.20 | 57.31 | 73.91 |
| | (±2.36) | (±1.41) | (±0.77) | (±3.25) | (±1.13) | (±0.47) | (±0.71) | (±0.57) |
| HAN | 15.29 | 16.95 | 6.98 | 14.73 | 70.57 | 87.89 | - | - |
| | (±2.87) | (±2.71) | (±0.58) | (±1.69) | (±1.58) | (±0.62) | - | - |
| HGT | 11.98 | 20.12 | 8.03 | 17.79 | 77.24 | 89.63 | 64.01 | 81.54 |
| | (±2.23) | (±3.89) | (±0.87) | (±0.83) | (±3.50) | (±0.84) | (±0.36) | (±0.56) |
| HeCo | 10.32 | 18.01 | 10.78 | 15.26 | 65.04 | 83.29 | 53.13 | 71.81 |
| | (±1.12) | (±0.87) | (±0.41) | (±1.52) | (±1.26) | (±0.72) | (±0.47) | (±0.35) |
| SHGP | 10.80 | 19.28 | 11.34 | 17.44 | 68.22 | 85.34 | 54.49 | 72.58 |
| | (±3.03) | (±0.91) | (±1.29) | (±1.49) | (±2.71) | (±0.48) | (±0.33) | (±0.24) |
| LM (XRoBERTa) | 40.10 | 44.71 | 54.51 | 61.27 | 60.20 | 84.23 | 51.21 | 74.22 |
| | (±4.62) | (±3.68) | (±1.50) | (±1.22) | (±2.78) | (±1.71) | (±0.17) | (±0.26) |
| LM (GPT-2) | 59.43 | 61.53 | 70.26 | 72.67 | 51.71 | 80.54 | 50.66 | 72.36 |
| | (±4.73) | (±3.43) | (±1.43) | (±0.90) | (±3.67) | (±2.49) | (±0.74) | (±0.86) |
| LM (DRoBERTa) | 58.29 | 60.57 | 66.25 | 70.14 | 60.97 | 83.00 | 51.44 | 75.09 |
| | (±2.44) | (±2.11) | (±1.60) | (±1.52) | (±2.98) | (±0.40) | (±0.14) | (±0.29) |
| LM +RGCN | 13.83 | 22.70 | 14.32 | 24.59 | 72.35 | 88.86 | 58.62 | 78.78 |
| | (±0.73) | (±3.25) | (±0.87) | (±1.17) | (±4.34) | (±1.46) | (±0.50) | (±0.10) |
| LM +HGT | 12.81 | 21.79 | 10.49 | 20.57 | 82.97 | 89.98 | 65.01 | 82.28 |
| | (±1.22) | (±3.54) | (±0.41) | (±0.97) | (±3.91) | (±0.88) | (±0.20) | (±0.30) |
| WalkLM | **60.42*** | **62.33*** | **75.16*** | **77.89*** | **85.65*** | **94.16*** | **82.15*** | **92.78*** |
| | (±2.62) | (±3.13) | (±0.93) | (±0.70) | (±3.28) | (±0.37) | (±0.67) | (±0.68) |

training tasks jointly based on a target set of relations to learn node embeddings and meta-paths on HINs.

(2) Relation learning-based methods: **ConvE** [6] proposes to use 2D convolution over embeddings and multiple layers of non-linear features to model KGs. **ComplEx** [49] handles a large number of binary relations using complex-valued embeddings on KGs. **SimKGC** [52] proposes to elicit the implicitly stored knowledge from BERT and designs a text-based contrastive learning mechanism for knowledge graph completion.

(3) Supervised heterogeneous graph neural networks (HGNNs): **RGCN** [45] proposes to apply GCN to model HINs or KGs. **HAN** [53] proposes to learn the importance between a node and its meta-path based neighbors on HINs. **HGT** [20] proposes to use each edge's type to parameterize the transformer-based self-attention architecture on HINs. For the above supervised HGNNs, we use link prediction loss introduced in GraphSAGE [14] to achieve unsupervised learning (i.e., without any node labels), following existing studies on HINs [11, 61].

(4) Unsupervised HGNNs: **HeCo** [54] proposes a co-contrastive learning mechanism for HGNNs. **SHGP** [63] designs a self-supervised pre-training method for HGNNs.

**Settings.** We mainly compare ten algorithms under the setting of unsupervised graph learning. The full code for this work is available[7]. All the models are optimized through the Adam optimizer and

---

[7]https://github.com/Melinda315/WalkLM

Table 3: Node classification results (%) in the few-shot setting with Macro-F1 (abbr. Ma-F1) and Micro-F1 (abbr. Mi-F1) metrics.

| Dataset | PubMed | | | | | | MIMIC-III | | | | | |
|---|---|---|---|---|---|---|---|---|---|---|---|---|
| Setting | 1 shot | | 3 shot | | 5 shot | | 1 shot | | 3 shot | | 5 shot | |
| Metric | Ma-F1 | Mi-F1 | Ma-F1 | Mi-F1 | Ma-F1 | Mi-F1 | Ma-F1 | Mi-F1 | Ma-F1 | Mi-F1 | Ma-F1 | Mi-F1 |
| ComplEx | 9.31 | 12.51 | 10.32 | 13.26 | 10.12 | 15.94 | 2.82 | 5.29 | 2.00 | 3.03 | 3.87 | 9.26 |
| M2V | 9.86 | 13.42 | 10.27 | 12.56 | 12.97 | 14.98 | 5.83 | 8.72 | 3.91 | 5.11 | 3.40 | 4.49 |
| ConvE | 13.23 | 13.45 | 8.84 | 10.93 | 11.25 | 13.53 | 5.85 | 6.75 | 5.61 | 7.24 | 6.31 | 7.69 |
| RGCN | 9.34 | 11.02 | 8.57 | 10.58 | 10.84 | 13.43 | 4.97 | 5.82 | 5.43 | 6.28 | 5.22 | 5.73 |
| HIN2Vec | 8.46 | 10.54 | 9.04 | 12.79 | 10.96 | 17.39 | 5.72 | 10.33 | 4.90 | 5.72 | 3.57 | 4.91 |
| SHGP | 8.94 | 12.79 | 9.12 | 11.73 | 10.53 | 15.14 | 4.12 | 6.35 | 5.36 | 6.58 | 4.47 | 5.34 |
| WalkLM | **28.09*** | **30.94*** | **32.11*** | **35.35*** | **35.41*** | **37.68*** | **23.33*** | **27.96*** | **34.19*** | **40.49*** | **41.12*** | **46.83*** |

the learning rate is searched in [1e-4, 1e-2]. The hyper-parameters of baselines are chosen carefully based on either grid search or their official source codes. For all the methods, we use a five-fold cross-validation for a more reliable evaluation of the model's performance. All the experiments are performed with two NVIDIA GTX 3090 Ti GPUs. For link prediction, we deploy the Large Margin Nearest Neighbor (LMNN) technique based on the embeddings generated by our WalkLM. Then we construct feature vectors for edges.

## 5.2 Node Classification

**Main Results.** For node classification, we train a separate one-layer MLP classifier based on the learned embeddings on 80% of the labeled nodes and predict the remaining 20%. All the methods are trained in an unsupervised manner without classification labels. We evaluate WalkLM with Macro-F1 (across all labels) and Micro-F1 (across all nodes).

As shown in Table 2, our proposed WalkLM has superior performance, indicating the importance of leveraging both semantic and structural information in attributed graphs. WalkLM achieves 138.59% performance gains on PubMed over the second-best performance on average while achieving 39.37% average performance gains on MIMIC-III. Specifically, SimKGC achieves second-best performance by effectively employing text-based contrastive learning, which leverages BERT to capture a rich set of semantic information. Compared with SimKGC, WalkLM can effectively combine the complex semantic and graph structure information of attributed graphs, so as to accurately model the complex attributes of nodes. Although the HGNNs can naturally model attributes, their unsupervised training mechanisms likely do not align well with the downstream prediction task of node classification.

**Results in the Few-shot Setting.** Since one key challenge of node classification lies in the generalizability and adaptability of models [63], we design a few-shot setting to evaluate models in extending knowledge to unseen scenarios and adapting to new tasks with limited training data.

As shown in Table 3, our framework can stay strong with a small size of training data, where we win a 275.45% performance gain over the second-best performance on average. Through the novel textualization process that converts general attributed graphs into text-like sequence data, our proposed WalkLM can leverage the capabilities of modern language models for graph representation learning. With the extensive pre-training of LMs on broad text corpora, WalkLM can easily understand meaningful node attributes given a new graph, while the random walk strategy further allows it to capture graph structures. Consequently, WalkLM maintains superior performance even in the few-shot setting. Similar to the main results in Table 2, the ranking of baselines is fluctuating across datasets, where ConvE and M2V continue to exhibit promising performance. Note that ComplEx and HIN2Vec exhibit notable improvements in this setting, likely because HIN2Vec can also learn node representations based on random walks and ComplEx can capture fine interactions through complex-valued vectors, thus being more capable of capturing comprehensive node information before supervision.

## 5.3 Link Prediction

For link prediction, we train all models with the randomly selected 80% links and evaluate towards the 20% held-out links. We use the Hadamard function to construct feature vectors for node pairs and train a two-layer MLP classifier on the 80% training links. We evaluate WalkLM with AUC (area under the ROC curve) and MRR (mean reciprocal rank). Note that, HAN cannot predict links on MIMIC-III for its restriction to embed only one type of node at a time, and thus it cannot predict links between different types of nodes on MIMIC-III [53].

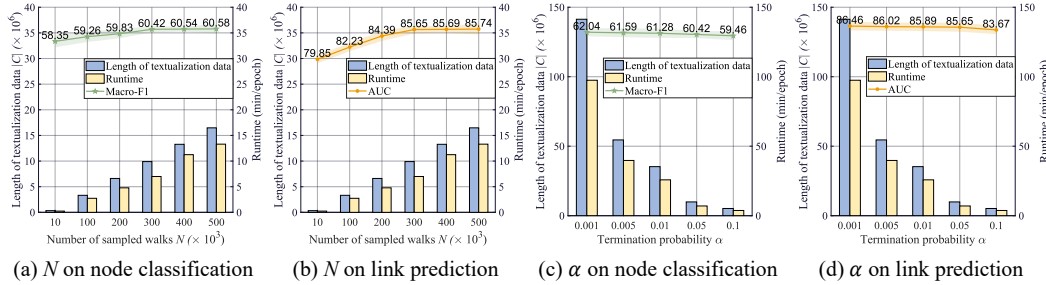

| (a) $N$ on node classification | (b) $N$ on link prediction | (c) $\alpha$ on node classification | (d) $\alpha$ on link prediction |

Figure 3: Analysis of the number of sampled walks $N$ and the termination probability $\alpha$.

As shown in Table 2, our fine-tuned `WalkLM` demonstrates outstanding performance in uncovering latent associations among nodes in attributed graphs. In general, `WalkLM` outperforms all ten baselines with an average of 5.97% performance gain over the second-best performance, showing that our proposed framework can learn accurate edge representation for link prediction. As a RW-based method, M2V can effectively employ meta-path-guided random walks to capture topological information and trace meta-path to understand relations between nodes. As relation-learning methods, ConvE and ComplEx design different deep neural models to evaluate triplets. ConvE can achieve good performance by using convolutions over embeddings to mine relations between entities. ComplEx can capture relations via complex-valued embeddings, so as to better represent the complex inherent relations among entities. Compare to M2V, ConvE, and ComplEX, `WalkLM` can effectively capture the complex relations by providing characteristic graph traits and reconstructing network proximity of nodes that inherit from RWs. On the other hand, the unsupervised HGNNs, especially HeCo and SHGP, perform rather poorly, again because their training mechanisms are not aligned with the link prediction task. Such observation is consistent with the results in the recent work [72], showing the heterogeneous approaches that only preserve certain-type entities fail to capture accurate representations for all kinds of nodes.

## 5.4 Ablation Studies

To better understand our proposed techniques, we closely study our framework by selecting different LMs and varying the graph-aware LM fine-tuning mechanism.

Compared with the graph-based baselines, the LM-based models (e.g., LM (XRoBERTa[8]), LM (GPT-2[9]), and LM (DRoBERTa[10])) are able to learn accurate and rich node attributes. However, it is difficult for them to mine the relations between nodes in the attributed graph, where all of them perform worse than the RW-based and relation learning-based methods in the link prediction task. Considering the overall performance of the above three LMs on two different tasks and the goal of learning graph embedding, we choose LM (DRoBERTa) as our starting point for fine-tuning.

Furthermore, we show that LM can further effectively integrate with existing heterogeneous graph algorithms, such as LM + RGCN and LM + HGT, resulting in a notable performance enhancement over their individual methods. Compared with LM + RGCN and LM + HGT, our proposed graph-aware LM fine-tuning can achieve the largest improvement gains based on the chosen LM (DRoBERTa) in both node classification and link prediction tasks, showing the effectiveness of capturing topological information together with semantics in modeling attributed graphs. The detailed analysis of the ablation studies is shown in Appendix A.3.

## 5.5 Hyper-parameter Studies

In this subsection, we investigate the model sensitivity on the number of sampled walks $N$ and the termination probability $\alpha$, which are the major hyper-parameters in `WalkLM`. For the space limitation, we show results on PubMed in Figure 3 and the results on MIMIC-III in Appendix A.4. Overall, `WalkLM` is not sensitive to the two hyper-parameters, where its performance increases slowly with $N$ and $\alpha$. Note that, too small $N$ or large $\alpha$ can cause the textualization data $|C|$ to lose

---

[8]https://github.com/facebookresearch/fairseq/tree/main/examples/xlmr

[9]https://github.com/openai/gpt-2

[10]https://github.com/huggingface/transformers/tree/main/examples/research_projects/distillation

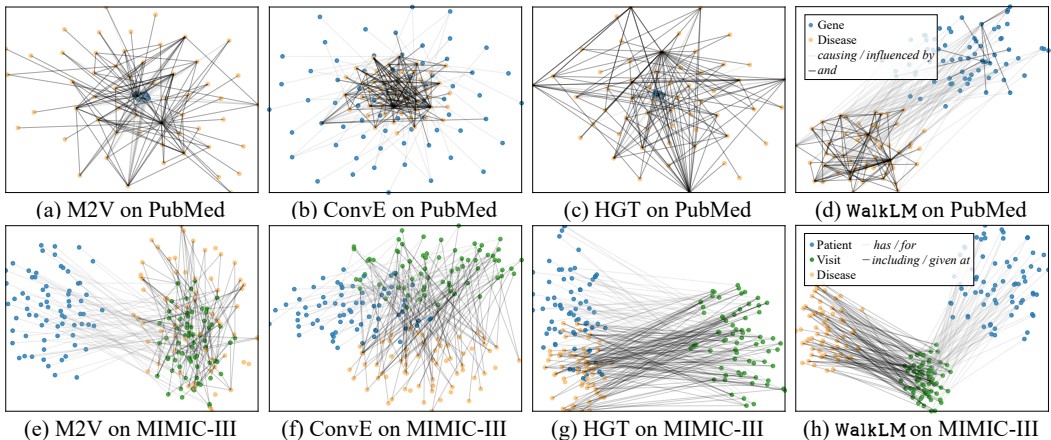

| (a) M2V on PubMed | (b) ConvE on PubMed | (c) HGT on PubMed | (d) WalkLM on PubMed |

| (e) M2V on MIMIC-III | (f) ConvE on MIMIC-III | (g) HGT on MIMIC-III | (h) WalkLM on MIMIC-III |

Figure 4: Visualization of different types of node embeddings on PubMed and MIMIC-III.

sufficient information, while too large $N$ or small $\alpha$ lead to extensive $|C|$ and increase computational costs for fine-tuning. Setting $N$ around $3 \times 10^5$ and $\alpha$ around 0.05 seems appropriate to generate sufficient textual sequences, which can achieve a good balance of performance and efficiency. We additionally investigate the model's sensitivity to the quantity of language model masking samples, aiming to elucidate the parameter's influence on the performance of downstream tasks. The detailed experimental results are shown in Appendix A.4.

## 5.6 Visualization

For an intuitive comparison, we visualize the embedding space of different types of nodes which are learned by M2V, ConvE, HGT, and our WalkLM, respectively. Specifically, we select gene/disease nodes on PubMed and patient/visit/disease nodes on MIMIC-III. The embeddings are further transformed into the 2-dimensional Euclidean space via the t-SNE algorithm [50]. The nodes and links are both colored according to their types.

As shown in Figure 4, M2V, ConvE, and HGT have blurred boundaries and even overlaps between different types of nodes, which are hard to distinguish. Our WalkLM shows the clearest boundaries between different types of nodes and the best within-type compactness, which indicate it can automatically organize heterogeneous nodes in a uniform space. Moreover, by connecting different types of nodes according to the relations in the data, we find that WalkLM can provide more discriminate distributions for different types of relations than others. The visualizations clearly demonstrate the advantages of WalkLM in capturing both attribute semantics and topological structures on graphs.

## 6 Conclusion

In this paper, we propose a novel uniform language model fine-tuning framework for attributed graph embedding. The proposed WalkLM consists of two key modules, which encapsulate both attribute semantics and graph structures and obtain unsupervised generic graph representations. We conduct extensive experiments to demonstrate the superior effectiveness of WalkLM against state-of-the-art baselines. For future work, it is intriguing to further design more sophisticated techniques and empirical evaluations toward the leverage of LMs and generalize our work to modern LLMs.

## 7 Acknowledgments

This work was supported in part by the National Natural Science Foundation of China (No. 6230071268); Fujian Provincial Youth Education and Scientific Research Project under Grant JAT220811. Carl Yang was not supported by any fund from China.

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

# A    Appendix

## A.1    Experiments based on KG datasets

**Datasets.** We conduct extensive experiments on two new real-world KG datasets, i.e., Freebase [11] and FB15K-237 [12]. Freebase contains a graph of books, films, sports, and locations. The nodes and edges are extracted according to [61]. A large portion of books are labeled into eight genres of literature. Each labeled book has only one label. FB15K-237 is a standard dataset in the knowledge graph community, which contains 310,116 triples with 14,541 entities and 237 relation types. Since we did not manually label the nodes, we only predicted whether a triple is correct or not on this dataset. We matched the entities with Wikidata entities and obtained metadata from Wikidata, and constructed a rough attribute graph dataset by using the names and descriptions of the nodes as textualized features of the nodes, and directly applying their original relationship text as the edge textualized attributes.

**Node Classification.** As shown in Table 4, our proposed `WalkLM` has superior performance, indicating the importance of leveraging both semantic and structural information in attributed graphs. `WalkLM` achieves 40.24% performance gains on Freebase over the second-best performance on average. Specifically, as a text-based Knowledge graph completion method, SimKGC can effectively employ text-based contrastive learning to capture a rich set of semantic information. Compared with SimKGC, `WalkLM` can effectively combine the complex semantic and graph structure information of attributed graphs, so as to accurately model the complex attributes of nodes.

**Link Prediction.** We evaluate `WalkLM` with AUC and MRR. As shown in Table 4, our fine-tuned `WalkLM` demonstrates outstanding performance in uncovering latent associations among nodes in attributed graphs. In general, `WalkLM` outperforms all ten baselines with an average of 2.05% performance gain over the second-best performance, showing that our proposed framework can learn accurate edge representation for link prediction. ComplEx and ConvE consistently demonstrate promising performance by effectively capturing generic node representations. However, as a text-based Knowledge graph completion method, SimKGC can sometimes outperform others in terms of the MRR metric, where SimKGC can enhance semantic similarity between nodes through contrastive learning based on bi-encoder architecture and three types of negatives. Compared with ConvE, ComplEX, and SimKGC, `WalkLM` can effectively capture the complex relations by providing text-based semantic traits of characteristic graph and reconstructing network proximity of nodes that inherit from RWs.

## A.2    Graph-level Classification

Compared with node or edge classification, aggregating node embeddings for graph-level classification needs more context information. Furthermore, graph-level classification presents its own set of challenges, which require holistic capturing of graph structures and often do not rely much on attributes. Therefore, it is difficult to find a universal representation learning approach that solves all different levels of graph mining tasks. Technically, adapting our method to graph-level classification necessitates some subtle decisions to make (such as whether to include graph ID as a virtual node). We've conducted a preliminary analysis on aggregating our learned node embeddings for graph-level tasks. Specifically, we adopt a widely-used MUTAG [13] dataset and use mean accuracy as the metric [48, 71]. The results on the popular MUTAG dataset are listed in Table 5. Although the findings are encouraging and show the potential of `WalkLM`, further studies are still needed to establish a clear advantage of our approach over SOTA graph classification baselines.

## A.3    Detailed Ablation Studies

From Table 6, we have the following observations: (1) Compared with the graph-based baselines, the LM-based models (e.g., LM (XRoBERTa), LM (GPT-2), and LM (DRoBERTa)) are able to learn accurate and rich node attributes, leading to superior performance in node classification. For the PubMed dataset distributed on 8 classes, LM (XRoBERTa), LM (GPT-2), and LM (DRoBERTa) achieve 63.78%, 135.04%, and 130.89% performance gains over the second-best performance on

---

[11]http://www.freebase.com/

[12]https://paperswithcode.com/dataset/fb15k-237

[13]https://ls11-www.cs.tu-dortmund.de/staff/morris/graphkerneldatasets

Table 4: Different downstream task results (%) with the corresponding std (±) on two KG datasets. The best performances are in bold and the second runners are shaded in gray, where * denotes a significant improvement according to the Wilcoxon signed-rank significance test.

| Task | Node Classification | | Link Prediction | | | |
|---|---|---|---|---|---|---|
| Dataset | Freebase | | Freebase | | FB15K-237 | |
| Metric | Macro-F1 | Micro-F1 | AUC | MRR | AUC | MRR |
| M2V | 25.74±1.12 | 50.25±2.57 | 80.68±1.81 | 88.97±0.93 | 90.35±0.50 | 96.78±0.19 |
| HIN2Vec | 15.56±1.07 | 43.67±2.12 | 80.04±3.01 | 90.90 ±0.87 | 79.68±0.83 | 92.85±0.40 |
| ConvE | 25.13±1.83 | 49.31±3.45 | 88.14±1.03 | 93.57±0.42 | 92.88±0.42 | 97.57±0.15 |
| ComplEx | 20.25±1.62 | 49.43±3.57 | 84.01±1.43 | 91.46±0.56 | 95.03±0.35 | 97.88±0.22 |
| SimKGC | 35.88±0.87 | 56.12±0.45 | 87.33±1.51 | 94.21±0.34 | 93.80±0.31 | 97.62±0.30 |
| RGCN | 15.37±1.54 | 45.86±1.03 | 82.75±0.89 | 91.52±0.64 | 85.88±0.35 | 89.84±0.19 |
| HAN | 14.25±1.77 | 39.30±2.18 | 80.73±1.37 | 91.61±0.34 | 82.06±0.53 | 89.31±0.89 |
| HGT | 19.97±1.34 | 47.99±2.56 | 81.94±1.84 | 89.65±0.43 | 87.41±0.69 | 94.62±0.34 |
| HeCo | 23.95±1.45 | 48.62±1.13 | 79.32±0.86 | 87.40±0.32 | 78.82±0.37 | 90.41±0.23 |
| SHGP | 13.83±1.27 | 39.07±1.39 | 78.37±1.77 | 85.52±0.69 | 78.56±0.33 | 89.84±0.21 |
| XRoBERTa | 48.10±2.01 | 67.95±0.97 | 73.94±1.62 | 88.17±0.91 | 75.62±0.72 | 91.10±0.71 |
| GPT-2 | 49.24±2.12 | 68.28±1.37 | 60.45±2.43 | 83.29±1.87 | 68.87±1.21 | 85.23±1.73 |
| DRoBERTa | 51.76±1.24 | 69.51±0.73 | 79.22±1.85 | 91.21±1.17 | 84.15±0.63 | 93.39±0.39 |
| LM+RGCN | 28.38±0.63 | 53.37±2.27 | 83.63±1.81 | 96.38±0.67 | 87.72±0.50 | 94.47±0.46 |
| LM+HGT | 20.79±0.67 | 48.73±3.13 | 83.09±1.23 | 89.79±0.35 | 88.18±0.61 | 94.85±0.27 |
| WalkLM | **55.01±2.67\*** | **71.36±1.53\*** | **92.11±2.24\*** | **96.54±0.56\*** | **95.65±0.18\*** | **98.45±0.33\*** |

Table 5: Accuracy results (%) of graph-level classification on MUTAG.

| Dataset | MUTAG | | | | | |
|---|---|---|---|---|---|---|
| Model | HIN2Vec | ConvE | ComplEx | LM (DRoBERTa) | WalkLM w/o. graph-ID | WalkLM |
| Accuracy | 78.72 | 77.64 | 78.69 | 79.23 | 79.77 | **81.39\*** |

average, respectively. For the MIMIC-III dataset on the total 19 classes, LM (XRoBERTa), LM (GPT-2), and LM (DRoBERTa) achieve 5.17%, 30.17%, and 20.12% average performance gains compared to the second-best performance, respectively.

(2) The choice of LMs can affect the performance of fine-tuning. Due to different pre-training corpora, LM (XRoBERTa) performs worse than LM (DRoBERTa) in most cases. Moreover, LM (GPT-2) achieves an average of 3.30% improvement over LM (DRoBERTa) in node classification, while LM (DRoBERTa) achieves an average of 7.08% improvement over LM (GPT-2) in link prediction. Considering the overall performance of the above three LMs on two different tasks and the goal of learning graph embedding, we choose LM (DRoBERTa) as our starting point for fine-tuning.

(3) Furthermore, LM can further effectively integrate with existing heterogeneous graph algorithms, resulting in a notable performance enhancement over their individual methods. Specifically, compared with RGCN, LM + RGCN achieves an average of 50.38% improvement in node classification, and achieves up to 6.59% improvements in link prediction. Compared with HGT, LM + HGT achieves up to 30.64% improvements in node classification and 7.42% improvements in link prediction.

(4) Compared with LM + RGCN and LM + HGT, our proposed graph-aware LM fine-tuning can achieve the largest improvement gains based on the chosen LM (DRoBERTa) in both node classification and link prediction tasks, showing the effectiveness of capturing topological information together with semantics in modeling attributed graphs. Specifically, our WalkLM outperforms the chosen LM (DRoBERTa) by up to 13.45% in node classification. In Particular, our WalkLM achieves up to 59.70% improvements in link prediction, which demonstrates our WalkLM can better learn accurate edge representation for link prediction by the graph-aware LM fine-tuning.

## A.4 Detailed Hyper-parameter Studies

We show the results of the model sensitivity on the number of sampled walks $N$ and the termination probability $\alpha$ on MIMIC-III in Figure 5. Overall, WalkLM is not sensitive to the two hyper-parameters, where its performance increases slowly with $N$ and $\alpha$. Note that, setting $N$ around $3 \times 10^5$ and $\alpha$ around 0.05 seems appropriate to generate sufficient textual sequences and limit computational costs

Table 6: The detailed ablation results (%) with the corresponding std (±) on two datasets. The best performances are in bold and the second runners are shaded in gray, where * denotes a significant improvement according to the Wilcoxon signed-rank significance test.

| Task | Node Classification | | | | Link Prediction | | | |
|---|---|---|---|---|---|---|---|---|
| Dataset | PubMed | | MIMIC-III | | PubMed | | MIMIC-III | |
| Metric | Macro-F1 | Micro-F1 | Macro-F1 | Micro-F1 | AUC | MRR | AUC | MRR |
| M2V | 15.35 | 20.27 | 19.69 | 29.24 | 74.53 | 89.58 | 75.05 | 88.32 |
| | (±1.27) | (±3.01) | (±0.62) | (±1.57) | (±3.79) | (±2.05) | (±0.41) | (±0.23) |
| HIN2Vec | 11.57 | 18.92 | 19.12 | 28.05 | 74.21 | 90.56 | 73.46 | 88.10 |
| | (±1.23) | (±2.78) | (±1.32) | (±1.44) | (±5.49) | (±1.06) | (±0.41) | (±0.14) |
| ConvE | 16.06 | 19.16 | 24.44 | 32.89 | 76.48 | 92.27 | 69.56 | 84.88 |
| | (±3.69) | (±4.00) | (±1.28) | (±0.86) | (±4.31) | (±0.57) | (±0.36) | (±0.25) |
| ComplEx | 13.93 | 18.27 | 9.82 | 21.39 | 79.81 | 91.79 | 63.86 | 81.40 |
| | (±2.59) | (±4.12) | (±0.56) | (±3.12) | (±0.97) | (±0.48) | (±0.42) | (±0.40) |
| SimKGC | 21.97 | 30.83 | 51.62 | 58.50 | 79.62 | 91.43 | 67.73 | 84.86 |
| | (±3.51) | (±3.10) | (±1.81) | (±1.52) | (±2.72) | (±0.48) | (±1.69) | (±0.54) |
| RGCN | 12.50 | 18.50 | 7.19 | 14.55 | 72.08 | 88.20 | 57.31 | 73.91 |
| | (±2.36) | (±1.41) | (±0.77) | (±3.25) | (±1.13) | (±0.47) | (±0.71) | (±0.57) |
| HAN | 15.29 | 16.95 | 6.98 | 14.73 | 70.57 | 87.89 | - | - |
| | (±2.87) | (±2.71) | (±0.58) | (±1.69) | (±1.58) | (±0.62) | - | - |
| HGT | 11.98 | 20.12 | 8.03 | 17.79 | 77.24 | 89.63 | 64.01 | 81.54 |
| | (±2.23) | (±3.89) | (±0.87) | (±0.83) | (±3.50) | (±0.84) | (±0.36) | (±0.56) |
| HeCo | 10.32 | 18.01 | 10.78 | 15.26 | 65.04 | 83.29 | 53.13 | 71.81 |
| | (±1.12) | (±0.87) | (±0.41) | (±1.52) | (±1.26) | (±0.72) | (±0.47) | (±0.35) |
| SHGP | 10.80 | 19.28 | 11.34 | 17.44 | 68.22 | 85.34 | 54.49 | 72.58 |
| | (±3.03) | (±0.91) | (±1.29) | (±1.49) | (±2.71) | (±0.48) | (±0.33) | (±0.24) |
| LM (XRoBERTa) | 40.10 | 44.71 | 54.51 | 61.27 | 60.20 | 84.23 | 51.21 | 74.22 |
| | (±4.62) | (±3.68) | (±1.50) | (±1.22) | (±2.78) | (±1.71) | (±0.17) | (±0.26) |
| LM (GPT-2) | 59.43 | 61.53 | 70.26 | 72.67 | 51.71 | 80.54 | 50.66 | 72.36 |
| | (±4.73) | (±3.43) | (±1.43) | (±0.90) | (±3.67) | (±2.49) | (±0.74) | (±0.86) |
| LM (DRoBERTa) | 58.29 | 60.57 | 66.25 | 70.14 | 60.97 | 83.00 | 51.44 | 75.09 |
| | (±2.44) | (±2.11) | (±1.60) | (±1.52) | (±2.98) | (±0.40) | (±0.14) | (±0.29) |
| LM +RGCN | 13.83 | 22.70 | 14.32 | 24.59 | 72.35 | 88.86 | 58.62 | 78.78 |
| | (±0.73) | (±3.25) | (±0.87) | (±1.17) | (±4.34) | (±1.46) | (±0.50) | (±0.10) |
| LM +HGT | 12.81 | 21.79 | 10.49 | 20.57 | 82.97 | 89.98 | 65.01 | 82.28 |
| | (±1.22) | (±3.54) | (±0.41) | (±0.97) | (±3.91) | (±0.88) | (±0.20) | (±0.30) |
| WalkLM | **60.42*** | **62.33*** | **75.16*** | **77.89*** | **85.65*** | **94.16*** | **82.15*** | **92.78*** |
| | (±2.62) | (±3.13) | (±0.93) | (±0.70) | (±3.28) | (±0.37) | (±0.67) | (±0.68) |

Table 7: Different downstream task results (%) with ratio of masked samples **m** on PubMed.

| Task | Node Classification | | Link Prediction | |
|---|---|---|---|---|
| Metric | Macro-F1 | Micro-F1 | AUC | MRR |
| **m** = 0.05 | 52.97 | 56.33 | 83.16 | 93.47 |
| **m** = 0.15 | **60.42*** | **62.33*** | **85.65*** | **94.16*** |
| **m** = 0.25 | 53.80 | 56.09 | 82.92 | 93.75 |
| **m** = 0.35 | 52.22 | 55.61 | 82.38 | 92.72 |

for fine-tuning, which can achieve a good balance of performance and efficiency. Furthermore, for the ratio of masked samples **m**, the specific results are listed in Table 7. Overall, WalkLM is sensitive to **m**, where the optimal value across different tasks is 0.15, which is consistent with the empirical selection in our paper and the previous work [7, 31, 40, 60]

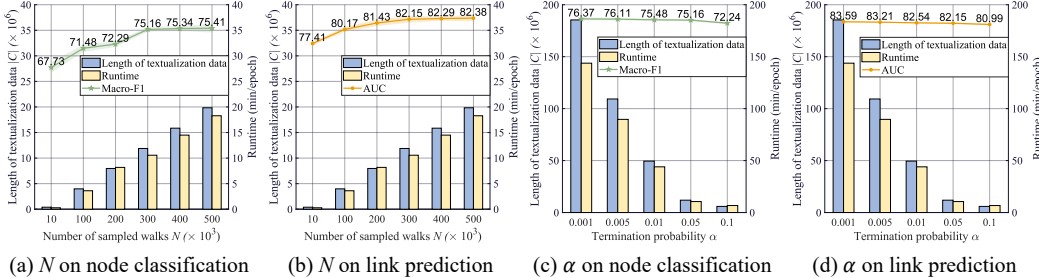

(a) $N$ on node classification    (b) $N$ on link prediction    (c) $\alpha$ on node classification    (d) $\alpha$ on link prediction

Figure 5: Analysis of the number of sampled walks $N$ and the termination probability $\alpha$.

