# OpenReview forum: "WalkLM: A Uniform Language Model Fine-tuning Framework for Attributed Graph Embedding"
_NeurIPS.cc/2023/Conference — NeurIPS 2023 poster_

### Official Review · Reviewer_DtxK · 2023-07-03

**Soundness:** 4 excellent
**Presentation:** 4 excellent
**Contribution:** 3 good
**Rating:** 8
**Confidence:** 4

**Summary:**

This paper studies the universal language model for generic graph representation learning. To model the complex attributes on multiple types of nodes and links with the consideration of graph structure, the authors proposed WalkLM. Specifically, to compose meaningful textual sequences, the attributed RWs and an automated program are exploited. Besides, to transfer the capability of language models, a fine-tuning strategy is designed to extract embedding vectors from the LM. Extensive experiments on real-world datasets show the superiority of proposed method over the state-of-the-art methods.

**Strengths:**

S1. This paper is well motivated. Considering the graph representations are generally limited to specific downstream predictions, and the performance may be unsatisfactory with unsupervised GNN ways, the authors integrate language models (LMs) and random walks (RWs) to obtain unsupervised generic graph representations

S2. The paper is well written and organized, making it easy to follow for readers and the work archives promising different downstream task results.

S3. The problem setting is quite realistic, and the experimental results are also encouraging. In my opinion, this work has potential to promote the application of general-purpose graph methods to more real-world problems.

S4. The adoption of random walk is a quite highlight part of this paper, which can easily capture flexible graph topological structures without supervision for various independent downstream tasks.


**Weaknesses:**

W1. It is suggested to include the deeper analysis on why the proposed framework can stay strong with a small size of training data for results in the few-shot setting.

W2. The analysis of the some hyperparameters need to be considered, where the hyperparameters may have an impact on the performance of the downstream tasks. For example, the latent dimension d and the number of masked samples.


**Questions:**

NA

**Limitations:**

See the Weaknesses part.

---

> ### Author Rebuttal · Authors · 2023-08-07
>
> We thank the reviewers for their positive and detailed comments, which affirm the importance of the problem we studied. In response to the weaknesses, our answers are as follows:
>
> > **W1:** It is suggested to include the deeper analysis on why the proposed framework can stay strong with a small size of training data for results in the few-shot setting.
>
> Through the novel textualization process that converts general attributed graphs into text-like sequence data, our proposed WalkLM can leverage the capabilities of modern language models for graph representation learning. With the extensive pre-training of LMs on broad text corpora, the model can easily understand meaningful node attributes given a new graph, while the random walk strategy further allows it to capture graph structures.  This is how WalkLM can consistently exhibit superior performance in the few-shot setting.
>
> > **W2:** The analysis of the some hyperparameters need to be considered, where the hyperparameters may have an impact on the performance of the downstream tasks. For example, the latent dimension d and the number of masked samples.
>
> We have analyzed specific hyperparameters in Section 5.5 of our paper (i.e., the number of sampled walks $N$ and the termination probability $\alpha$). Furthermore, we have added experiments on the suggested hyperparameters. Regarding the latent dimension $d$, it's a common practice to utilize the output dimension of the original LMs (e.g., the dimension in GPT is 768 [1]). For the ratio of masked samples $m$, the specific results are listed as follows, where the optimal value across different tasks is 0.15, which is consistent with the empirical selection in our paper and the previous work [2]-[4]:
>
> **Table 1: Different downstream task results (%) with varing $m$ on PubMed.**
> | Dataset      |                     | PubMed |                 |       |
> | ------------ | ------------------- | -------- | --------------- | ----- |
> |   Task   | Node Classification |           | Link Prediction |           |
> |  Metric  |      Macro-F1       | Micro-F1  |       AUC       |    MRR    |
> | $m=0.05$ |        52.97        |   56.33   |      83.16      |   93.47   |
> | $m=0.15$ |      **60.42**      | **62.34** |    **85.65**    | **94.16** |
> | $m=0.25$ |        53.80        |   56.09   |      82.92      |   93.75   |
> | $m=0.35$ |        52.22        |   55.61   |      82.38      |   92.72   |
>
> **Reference**
> [1] Improving language understanding by generative pre-training, 2018.
> [2] BERT: Pre-training of Deep Bidirectional Transformers for Language Understanding, Proceedings of naacL-HLT, 2019.
> [3] RoBERTa: A Robustly Optimized BERT Pretraining Approach, arXiv e-prints, 2019.
> [4] Clinical-BERT: Vision-Language Pre-training for Radiograph Diagnosis and Reports Generation, AAAI, 2022.

---

> > ### Comment · Reviewer_DtxK · 2023-08-16
> > **Response after rebuttal**
> >
> > The authors’ rebuttal addressed most of my concerns and I am happy to see the paper accepted (raising my score from 7 to 8).

---

> > > ### Author Response · Authors · 2023-08-17
> > > **Thanks for the response**
> > >
> > > Dear reviewer DtxK,
> > >
> > > We thank your response and appreciation of our work and rebuttal. We will make sure to incorporate the new results and discussions into our revision.
> > >
> > > Best,
> > > Authors

---

### Official Review · Reviewer_x99Y · 2023-07-04

**Soundness:** 3 good
**Presentation:** 3 good
**Contribution:** 3 good
**Rating:** 8
**Confidence:** 4

**Summary:**

GNNs for training require sufficient training data for downstream tasks to achieve strong performance. Self supervised learning approaches are inefficient due to the presence of a variety of node attributes and complicated relations between nodes. Inspired from the success in LLMs, they convert the graphs into natural language to be consumed by pre -trained language models. Random walk sequences are extracted from the graph and are converted into text through entity level and walklevel textualization. Roberta model is then fine tuned to predict the masked node / edges as textual sequences. The approach is tried on various node classification and link prediction tasks with significant improvements.

**Strengths:**

1. Since the main crux of the approach is how we textualize the graphs. The method is highly flexible to extract node, edge, sub-graph, path-specific or even graph embeddings.
2. I like how self-sufficient the paper is. All essential material to understand and analyze has been fitted into the main paper.
3. The technique has been compared to a lot of standard GNN approaches and in the ablation LLM + GNN have been combined to further enhance performance and use the best of both worlds.
4. Overall the approach is elegant and easy to understand. It brings huge improvements to downstream tasks. I think the approach is a significant contribution to the domain of LLMs and graphs.


**Weaknesses:**

1. The authors do not present results on the graph level classification task. Only node classification and link prediction tasks have been presented. It would be interesting to see if aggregating node embeddings for graph-level tasks also performs well as this task needs more context as compared to node or edge classification.

**Questions:**

1. Do the authors have an opinion on - applying the same technique for nontextual networks with nontextual Ids and numeric attributes? I think the bigger question is how well can transformer-like models embed graph structures without the necessity of using GNN-like graph embeddings.


**Limitations:**

1. The process of converting graphs to text as presented in the paper consists of entity level textualization which can be taxing as the number and types of nodes increase to millions.
2. Extracting random walks is an expensive operation and this method will have scalability issues when the amount of nodes/edges increases to millions or billions

---

> ### Author Rebuttal · Authors · 2023-08-07
>
> We thank the reviewer for the overall positive summary and accurate description of our major contributions. As for the several weaknesses you mentioned, our responses are listed as follows:
>
> > **W1:** The authors do not present results on the graph level classification task. Only node classification and link prediction tasks have been presented. It would be interesting to see if aggregating node embeddings for graph-level tasks also performs well as this task needs more context as compared to node or edge classification.
>
> We appreciate your constructive suggestion. Graph-level classification presents its own set of challenges, which require holistic capturing of graph structures and often do not rely much on attributes. Therefore, it is difficult to find a universal representation learning approach that solves all different levels of graph mining tasks. Technically, adapting our method to graph-level classification necessitates some subtle decisions to make (such as whether to include graph ID as a virtual node). Following the reviewer's suggestion, we've conducted a preliminary analysis on aggregating our learned node embeddings for graph-level tasks. Specifically, we adopt a widely-used MUTAG dataset and use mean accuracy as the metric [1][2]. The results on the popular MUTAG dataset are listed in Table 1. Although the findings are encouraging and show the potential of WalkLM, further studies are still needed to establish a clear advantage of our approach over SOTA graph classification baselines.
>
> **Table 1: Accuracy results (%) of graph-level classification on MUTAG.**
> | Dataset|      |  | |  MUTAG     |  |       |
> | :---: | :---: | :-----: | :----: | :----------: | :------------------: | :-------: |
> | Model | ConvE | ComplEx | HinVec | LM(DRoBERTa) | WalkLM w/o. graph-ID |  WalkLM   |
> |Accuracy | 77.64 |  78.69  | 78.72  |    79.23     |        79.77         | **81.39** |
>
> **Reference**
> [1] S2GAE: Self-Supervised Graph Autoencoders are Generalizable Learners with Graph Masking, WSDM, 2023.
> [2] An end-to-end deep learning architecture for graph classification, AAAI, 2018.
>
> > **Q1:** Do the authors have an opinion on - applying the same technique for nontextual networks with nontextual Ids and numeric attributes? I think the bigger question is how well can transformer-like models embed graph structures without the necessity of using GNN-like graph embeddings.
>
> Thanks for the question.  In this work, we apply RW to help LM in capturing graph structures. In our experiments, we have found that nontextual IDs can be effectively embedded as new tokens that appear in different random-based sequences. However, we have not yet found an effective way to model fully nontextual networks and numeric attributes. It is promising to establish our framework for more general graph representation along with the active research on more powerful and transparent LMs.
>
> > **L1:** The process of converting graphs to text as presented in the paper consists of entity level textualization which can be taxing as the number and types of nodes increase to millions.
>
> We only need to perform the rule-based textualization once for every node during the pre-processing stage, which is not only pretty fast but also highly amenable to parallelization.
>
> > **L2:** Extracting random walks is an expensive operation and this method will have scalability issues when the amount of nodes/edges increases to millions or billions.
>
> Instead of being a limitation, using random walks for capturing graph structures is in fact very efficient and scalable, which is an advantage of our method. Specifically, in industry, random walks on large graphs such as those with millions to billions of nodes can be done on CPUs with terabytes of memory with numerous threads in parallel [1].
>
> **Reference**
> [1] Pixie: A system for recommending 3+ billion items to 200+ million users in real-time, WWW, 2018.

---

### Official Review · Reviewer_4W5x · 2023-07-05

**Soundness:** 2 fair
**Presentation:** 3 good
**Contribution:** 2 fair
**Rating:** 4
**Confidence:** 4

**Summary:**

The paper proposed a method for knowledge-graph-embedding (KBE) task using integration of language model and random walks. Specifically, authors first verbalized the path via random walks in KB, then, fine-tuning language model for the verbalized path, finally, using the embedding layer of language model as the embedding of KB. Also, the numerical studies compared 9 different baseline graph embedding methods on downstream tasks node classification and link prediction. The proposed method WalkLM shows a good performance compared to listed baselines.

**Strengths:**

1. The paper is well organized and easy to follow.
2. The proposed method WalkLM shows a good performance compared to baselines.

**Weaknesses:**

1. My top concern is the novelty. The proposed method simply fine-tuned a language model based on path from random walk. However, similar ideas have been widely explored by previous papers like [1]-[4].
2. In section 2 related work, a lot of related papers about KB + language model besides [1]-[4] are not included.


[1] KEPLER: A Unified Model for Knowledge Embedding and Pre-trained Language Representation
[2] Exploiting Structured Knowledge in Text via Graph-Guided Representation Learning
[3] Deep Bidirectional Language-Knowledge Graph Pretraining
[4] Jaket: Joint pre-training of knowledge graph and language understanding

**Questions:**

It would be great to include more recent KB + language works as baseline to show the good performance of presented method.

---

> ### Author Rebuttal · Authors · 2023-08-07
>
> We thank the reviewer for the comments. Our responses are listed as follows:
>
> > **W1:** My top concern is the novelty. The proposed method simply fine-tuned a language model based on path from random walk. However, similar ideas have been widely explored by previous papers like [1]-[4].
>
> The setting and technical design of our method are new, where we leverage the capability of modern LMs for general attributed graph representation learning via a novel graph textualization process. The approach is concretely backed up by graph theory and properly leverages the advantages of random walks, which have been shown effective in capturing flexible graph topological structures. Note that, we do not intend to claim much novelty in the LM fine-tuning process. However, we believe that fine-tuning LMs for vertical or non-textual domain data and tasks can indeed lead to important discoveries, and many impactful contributions in the field have hinged on endeavors as such [1]-[5].
>
> **Reference**
> [1] Training language models to follow instructions with human feedback, NeurIPS, 2022.
> [2] Fine-tuning language models to find agreement among humans with diverse preferences, NeurIPS, 2022.
> [3] Generating training data with language models: Towards zero-shot language understanding, NeurIPS, 2022.
> [4]  Solving Math Word Problems via Cooperative Reasoning induced Language Models, ACL, 2023.
> [5] Language Models Get a Gender Makeover: Mitigating Gender Bias with Few-Shot Data Interventions, ACL, 2023.
>
>
> > **W2:** In section 2 related work, a lot of related papers about KB + language model besides [1]-[4] are not included.
>
> We thank the reviewer for highlighting the related work on KB+LM. We will add the discussion such as regarding [1]-[4] in our related work. However, we want to emphasize that, the goal of this work is to leverage the capability of modern LMs for general attributed graph representation learning, instead of (1) leveraging KB for enhancing LMs [1]-[5] or (2) leveraging LM for KB completion [6]-[9]. Therefore, we believe this work is novel and significantly different from the existing work on KB+LM, and we will further clarify this in our revision.
>
> **Reference**
> [1] KEPLER: A Unified Model for Knowledge Embedding and Pre-trained Language Representation, TACL, 2021.
> [2] Exploiting Structured Knowledge in Text via Graph-Guided Representation Learning, EMNLP, 2020.
> [3] Deep Bidirectional Language-Knowledge Graph Pretraining, NeurIPS, 2021.
> [4] Jaket: Joint pre-training of knowledge graph and language understanding, AAAI, 2022.
> [5] Knowledge Enhanced Contextual Word Representations, EMNLP-IJCNLP, 2019.
> [6] Fusing topology contexts and logical rules in language models for knowledge graph completion, Information Fusion, 2023.
> [7] Multi-task learning for knowledge graph completion with pre-trained language models, COLING, 2020.
> [8] From discrimination to generation: Knowledge graph completion with generative transformer, WWW, 2022.
> [9] Multilingual Knowledge Graph Completion from Pretrained Language Models with Knowledge Constraints, ACL, 2023.
>
> > **Q1:** It would be great to include more recent KB + language works as baseline to show the good performance of presented method.
>
> As we discussed in the response to W2, our focus in this work is to leverage LMs for general attributed graph representation learning, which cannot be adequately achieved by existing KB+LM works. Although some studies such as [1] can perform KB representation learning, they do not apply to general attributed graphs (KBs usually do not have attributes). Furthermore, as suggested by Reviewer vuBn, we have added one KB dataset (i.e., Freebase [2]) to enhance the experimental results, which shows the generalizability of our method to actual KBs.
>
> **Table1: Different downstream task results (%) on FreeBase.**
>
> | Dataset      |                     | FreeBase  |                 |           |
> | ------------ | ------------------- | --------- | --------------- | --------- |
> | Task         | Node Classification |           | Link Prediction |           |
> | Metric       | Macro-F1            | Micro-F1  | AUC             | MRR       |
> | M2V          | 25.74               | 50.25     | 80.68           | 88.97     |
> | HIN2Vec      | 15.56               | 43.67     | 80.04           | 90.90      |
> | ConvE        | 25.13               | 49.31     | 88.14           | 93.57     |
> | ComplEx      | 20.25               | 49.43     | 84.01           | 91.46     |
> | RGCN         | 15.37               | 45.86     | 82.75           | 91.52     |
> | HAN          | 14.25               | 39.30      | 80.73           | 91.61     |
> | HGT          | 19.97               | 47.99     | 81.94           | 89.65      |
> | HeCo         | 23.95               | 48.62     | 79.32           | 87.40      |
> | SHGP         | 13.83               | 39.07     | 78.37           | 85.52     |
> | LM(DRoBERTa) | 51.76               | 69.51     | 79.22           | 91.21     |
> | WalkLM       | **55.01**           | **71.36** | **92.11**       | **96.54** |
>
> **Reference**
> [1] KEPLER: A Unified Model for Knowledge Embedding and Pre-trained Language Representation, TACL, 2021.
> [2] Heterogeneous network representation learning: A unified framework with survey and benchmark, TKDE, 2020.

---

> > ### Comment · Reviewer_4W5x · 2023-08-16
> > **Thank you for the reply**
> >
> > Thank you for the follow up. The weakness of missing KB+LM method discussion/comparison is not well addressed. The difference between applying WalkLM to KB (like Freebase) and attributed graph is not clear. I would like to keep the rating as borderline reject.

---

> > > ### Author Response · Authors · 2023-08-16
> > > **Thanks for the response and some clarifications**
> > >
> > > We thank the reviewer again for the response. Although we have tried to make it clear in the rebuttal, here we want to emphasize again the two points raised by the reviewer above:
> > >
> > > - The goal of this work is to **innovatively apply LM for general attributed graph representation learning**. As we have discussed in the answer to W2 above (and we will incorporate discussions as such into the revision), all existing studies on KB+LM are about **using KB to enhance LM** and/or **using LM to improve KB completion/reasoning**, both of which are rather different from our work.
> > >
> > > - KB/Freebase can be regarded as one type of attributed graphs, with attributes as simple as the node names. Thus WalkLM is not really designed for KB/Freebase, but it can be easily applied to KB/Freebase and yield reasonable performance.
> > >
> > > We hope this can further assist the reviewer to properly understand our work.

---

> > > > ### Comment · Reviewer_4W5x · 2023-08-16
> > > >
> > > > Thanks for the reply. I do not agree with statement `KB/Freebase can be regarded as one type of attributed graphs, with attributes as simple as the node names`. My understanding is that attributed graphs is a simplified type of KB, the nodes in Freebase usually have many properties besides name.
> > > >
> > > > For the statement `all existing studies on KB+LM are about using KB to enhance LM and/or using LM to improve KB completion/reasoning`, I do not agree with this one neither. I think there are many works such as [1]-[4] listed in weakness are using LM for Knowledge Graph Embedding (KGE). So my concern is that the difference between "apply LM for general attributed graph representation learning" and "apply LM for knowledge graph embedding" is unclear and not well discussed.

---

> > > > > ### Author Response · Authors · 2023-08-16
> > > > > **Official comment by authors**
> > > > >
> > > > > Thanks again for the further comment.
> > > > >
> > > > > It looks like we have some disagreement on the definition of KB. However, KB has never been the target of WalkLM in this work, so the discussions of its definition and different types of KBs are well beyond the scope of this work.
> > > > >
> > > > > In this work, we develop WalkLM for attributed graphs. For those simple KBs that can be easily transformed into attributed graphs, such as the version of FreeBase we used in the rebuttal, we have shown that WalkLM can be easily applied and yield reasonable performance. For those more complicated KBs as pointed out by the reviewer that may not be easily transformed into attributed graphs, we can freely admit that WalkLM may or may not apply to them at the current stage.
> > > > >
> > > > > We thank the reviewer for pointing this out and we are happy to include a detailed discussion as such in the revision.

---

> > > > > > ### Comment · Reviewer_4W5x · 2023-08-17
> > > > > >
> > > > > > Thanks for the reply. Yes, the discussion about the definition of KB is beyond the scope of the work. I also agree that adding the discussion about KB + LM in the next version is easily addressed.
> > > > > >
> > > > > > Will increase my score to 5.

---

> > > > > > > ### Author Response · Authors · 2023-08-17
> > > > > > > **Official comment by authors**
> > > > > > >
> > > > > > > Dear reviewer 4W5x,
> > > > > > >
> > > > > > > We really appreciate the discussions with you and thanks again for your time in the review and discussion. We will make sure to properly incorporate the new results and discussions into our revision.
> > > > > > >
> > > > > > > Best,
> > > > > > > Authors

---

### Official Review · Reviewer_vuBn · 2023-07-07

**Soundness:** 3 good
**Presentation:** 3 good
**Contribution:** 3 good
**Rating:** 6
**Confidence:** 4

**Summary:**

This paper proposes WalkLM, an unsupervised graph representation learning leveraging the power of the language model. WalkLM first samples a set of sequences of entities from attributed graphs by random walk and fine-tunes the language model on textualized walks. The learned embedding by these procedures is employed to node-level and edge-level downstream tasks. The authors demonstrate that WalkLM significantly outperforms baselines in node classification and link prediction tasks on two real-world datasets.


**Strengths:**

WalkLM is a neat idea that combines graph representation learning and language modeling. This paper shows how the classic random walk approach can enhance graph representations using the expressiveness of the language model. The authors conduct various experiments including ablation studies, hyperparameter analysis, qualitative analysis, and efficiency analysis. There is no particular reason to reject the paper.


**Weaknesses:**

One weakness is the number of datasets used for evaluation. Enough experiments have been done, but only two datasets are covered in this paper. This paper proposed a new data format and corresponding training method beyond proposing a just new model architecture. So, it is necessary to clarify whether there are plenty of scenarios that this method can be easily used in the real world. What are the conditions of the tasks and datasets to which this method can be applied, and what datasets can be employed for WalkLM besides the two datasets? If possible, further experiments should be done on those datasets.


**Questions:**

See the weaknesses section.

---

> ### Author Rebuttal · Authors · 2023-08-07
>
> We thank the reviewer for the overall positive evaluations and detailed suggestions. In the following, we focus on the main issues to provide our feedback:
>
> > **1:** The conditions of the tasks and datasets to apply, and the number of datasets used for evaluation.
>
> The target of this work is general attributed graph representation learning, and the proposed WalkLM can learn node representations that can be used for various downstream tasks (e.g., disease prediction and topic modeling) and datasets (e.g., PubMed and MIMIC-III). In response to the reviewer's suggestions on adding more datasets for experiments, we have extended our experiments to include a dataset of FreeBase (as used in [1]). The results further confirm the generalizability of our proposed method:
>
> **Table1: Different downstream task results (%) on FreeBase.**
>
> | Dataset      |                     | FreeBase  |                 |           |
> | ------------ | ------------------- | --------- | --------------- | --------- |
> | Task         | Node Classification |           | Link Prediction |           |
> | Metric       | Macro-F1            | Micro-F1  | AUC             | MRR       |
> | M2V          | 25.74               | 50.25     | 80.68           | 88.97     |
> | HIN2Vec      | 15.56               | 43.67     | 80.04           | 90.90      |
> | ConvE        | 25.13               | 49.31     | 88.14           | 93.57     |
> | ComplEx      | 20.25               | 49.43     | 84.01           | 91.46     |
> | RGCN         | 15.37               | 45.86     | 82.75           | 91.52     |
> | HAN          | 14.25               | 39.30      | 80.73           | 91.61     |
> | HGT          | 19.97               | 47.99     | 81.94           | 89.65      |
> | HeCo         | 23.95               | 48.62     | 79.32           | 87.40      |
> | SHGP         | 13.83               | 39.07     | 78.37           | 85.52     |
> | LM(DRoBERTa) | 51.76               | 69.51     | 79.22           | 91.21     |
> | WalkLM       | **55.01**           | **71.36** | **92.11**       | **96.54** |
>
> **References**
> [1] Heterogeneous network representation learning: A unified framework with survey and benchmark, TKDE, 2020.
>
> > **2:** The new data format and the real-world applicability.
>
> Our method does not introduce a new data format, but rather introduces the novel process of *textualization*, which converts general attributed graphs into text-like sequence data. This process allows us to leverage the capabilities of pre-trained language models for graph representation learning. Importantly, our method only requires some meaningful attributes on the graphs, which is available in most real-world graphs such as biological networks, social networks, and knowledge graphs.

---

> > ### Comment · Reviewer_vuBn · 2023-08-16
> > **Response to the rebuttal**
> >
> > I acknowledged the author response. Thank you for addressing my questions. No further question is needed.

---

> > > ### Author Response · Authors · 2023-08-16
> > > **Thanks for the response**
> > >
> > > We thank the reviewer for the response and will make sure to include the new results and discussions in our rebuttal into the final version.

---

### Decision · Program_Chairs · 2023-09-21

**Decision:**

Accept (poster)

**Comment:**

If the authors have access to the review and discussion text of reviewer DtxK, that will greatly help them revise the paper to highlight its novelty and technical strength *way* better than they have done in the submission itself.

The authors should do three more things to justify the high scores, otherwise the paper may have limited appeal.

First, they should discuss all these prior papers below (in rough order of priority) and justify the novelty of their ideas against the backdrop of these papers.

 * Random Walks and Neural Network Language Models on Knowledge Bases. Josu Goikoetxea, Aitor Soroa and Eneko Agirre. NAACL 2015. "Each time the random walk reaches a vertex, a word is emitted at random using the probabilities in the inverse dictionary. When the random walk terminates, the sequence of emitted words forms the pseudo sentence which is fed to the NNLM architecture."
 * [DOLORES: Deep Contextualized Knowledge Graph Embeddings](https://arxiv.org/pdf/1811.00147.pdf). Haoyu Wang, Vivek Kulkarni, William Yang Wang. AKBC 2020.  (See "path generator" and "learner".)
 * [Exploring Large Language Models for Knowledge Graph Completion](https://www.semanticscholar.org/paper/Exploring-Large-Language-Models-for-Knowledge-Graph-Yao-Peng/9acc39255945b895faf0354b51716b227da87ea7) Liang Yao, Jiazhen Peng, Chengsheng Mao, Yuan Luo.
 * [KGBERT] KG-BERT: BERT for Knowledge Graph Completion. Liang Yao, Chengsheng Mao, Yuan Luo. 2019. "Our method takes entity and relation descriptions of a triple as input and computes scoring function of the triple with the KG-BERT language model."
 * Plus, include the discussion of the papers pointed out by reviewer 4W5x.


Second, they should add experiments involving the following data sets that are standard in the knowledge graph community, and compare against state-of-the-art KG inference systems.  For an accepted paper, this might be a tall ask, but it is my duty.
 * FB15K, FB15K237
 * WN18, WN18RR
 * YAGO3-10

Reviewer DtxK is of the opinion that this is not needed, because "attributed graphs" ("with both node and edge attributes") are allegedly different from KGs.  I cannot understand this claim, because certainly a KG like Wikidata has diverse attributes, both textual (in many languages!) and often structured, associated with nodes and edges.  It is certainly possible and desirable to know how much pretrained corpus information can boost purely structural completion algorithms on the above KGs, also because highly reliable standard numbers are available for them, unlike the two data sets used by the authors.


Third, there needs to be more diagnostics and explanations for why WalkLM is performing better than the closest algorithms it is compared against.